

# Image enhancement with art design: a visual feature approach with a CNN-transformer fusion model

Ming Xu[1], Jinwei Cui[1], Xiaoyu Ma[1], Zhiyi Zou[1], Zhisheng Xin[1] and Muhammad Bilal[2]

[1] School of Architecture and Art, Central South University, Changsha, China
[2] Department of Pharmaceutical Outcomes and Policy, University of Florida, Gainesville, FL, United States of America

Corresponding author
Xiaoyu Ma, 221301006@csu.edu.cn

## ABSTRACT

Graphic design, as a product of the burgeoning new media era, has seen its users' requirements for images continuously evolve. However, external factors such as light and noise often cause graphic design images to become distorted during acquisition. To enhance the definition of these images, this paper introduces a novel image enhancement model based on visual features. Initially, a histogram equalization (HE) algorithm is applied to enhance the graphic design images. Subsequently, image feature extraction is performed using a dual-flow network comprising convolutional neural network (CNN) and Transformer architectures. The CNN employs a residual dense block (RDB) to embed spatial local structure information with varying receptive fields. An improved attention mechanism module, attention feature fusion (AFF), is then introduced to integrate the image features extracted from the dual-flow network. Finally, through image perception quality guided adversarial learning, the model adjusts the initial enhanced image's color and recovers more details. Experimental results demonstrate that the proposed algorithm model achieves enhancement effects exceeding 90% on two large image datasets, which represents a 5%–10% improvement over other models. Furthermore, the algorithm exhibits superior performance in terms of peak signal-to-noise ratio (PSNR) and structural similarity index measure (SSIM) image quality evaluation metrics. Our findings indicate that the fusion model significantly enhances image quality, thereby advancing the field of graphic design and showcasing its potential in cultural and creative product design.

## INTRODUCTION

In recent years, with the continuous and rapid development of society, the pursuit of material wealth has gradually evolved into a quest for spiritual enrichment, encompassing personal development, interpersonal harmony, and an appreciation for truth, art, and beauty. Graphic design, to some extent, fulfills this pursuit of art and beauty, attracting increasing attention. Graphic design utilizes vision as a medium for communication and expression, creating and combining symbols, pictures, and words to convey ideas

or information visually (*Baine Breuna, 2021*; *Fosco et al., 2020*; *Wang et al., 2020*). This discipline not only provides a platform for professional designers but also creates more space for a broader array of art creators. However, the traditional methods of manual identification and extraction in graphic design have become inefficient, leading to issues such as inaccurate image classification and design bottlenecks as the application scope and resources expand. Image enhancement is a critical aspect of graphic design, as it directly influences the visual appeal, clarity, and impact of images used in various applications. High-quality images are essential for attracting attention, conveying messages effectively, and creating a professional and polished look. In fields such as advertising, branding, web design, and digital media, enhanced images can significantly improve user engagement and the overall effectiveness of visual communication. Despite its importance, image enhancement presents several challenges. Noise reduction is a significant issue, as removing noise without losing important details requires a delicate balance to avoid over-smoothing. Preserving fine details and textures while enhancing images is crucial to prevent them from looking artificial or overly processed. Accurate color correction is essential to maintain the natural appearance of images, particularly challenging under varied lighting conditions. Resolution enhancement, especially when converting low-resolution images to higher resolutions, must be done without introducing artifacts or blurriness. Furthermore, achieving high-quality enhancement with low latency is a common challenge in real-time processing applications such as video editing or live streaming.

The rapid advancement of artificial intelligence (AI) technology has significantly impacted various industries, including graphic design. AI can draw inspiration from a vast array of design works through deep learning algorithms, generating new design elements and compositions. This capability helps designers avoid over-reliance on traditional inspiration and experience during the creative process. Additionally, AI technology can automatically process images, enhancing design efficiency. Through AI algorithms, designers can quickly crop images, adjust colors, and add filter effects. This automation makes graphic design more convenient, saving time and energy for designers. In graphic design, layout and composition are crucial, and AI technology enables more precise typesetting, optimizing both the aesthetic appeal and readability of layouts. Despite these advancements, there are still many challenges in applying AI to graphic design, which can be addressed through image processing methods based on visual features.

Computer technology offers substantial advantages for image processing, leading to extensive research in this area. In the field of image enhancement, researchers have widely adopted deep learning methods, particularly utilizing single network models and generative adversarial network (GAN). While single models are convenient for processing data and operations, their effectiveness in image enhancement is relatively limited. GAN, on the other hand, achieve image generation through competitive training between generators and discriminators. The GAN framework comprises two neural networks: a generator, which creates realistic images, and a discriminator, which evaluates their authenticity. Integrating an attention mechanism within this framework allows the model to focus on different regions within the image, thereby enhancing image enhancement and synthesis.

These methods significantly improve the feasibility and effectiveness of graphic design image processing.

Building on this approach, we introduce a novel visual feature-based image enhancement model for graphic design, employing HE and image feature extraction through a dual-flow network combining convolutional neural network (CNN) and Transformer architectures. These methods are particularly suited for image enhancement tasks. CNNs are highly effective in extracting hierarchical features from images, making them ideal for tasks like denoising, super-resolution, and detail enhancement. By focusing on local regions of an image, CNNs can enhance fine details and textures without affecting the overall structure, preserving important visual elements. Transformers excel in capturing long-range dependencies and global context, crucial for tasks requiring an understanding of the entire image, such as color correction and contrast enhancement. The self-attention mechanism in transformers allows for adaptive focus on different parts of the image, enabling more nuanced and context-aware enhancements. AFF combines the strengths of different enhancement techniques by adaptively fusing features extracted by various methods, leveraging the detailed local features captured by CNNs and the global context provided by transformers. This approach allows for dynamic adjustment based on the specific requirements of the image enhancement task, resulting in more versatile and effective enhancement solutions. The specific contributions of this study are as follows:

(1) To enhance data utilization, we apply histogram equalization (HE) to improve image data by stretching the dynamic range of pixel gray value differences.

(2) To better identify graphic design images, we propose an image enhancement model based on a dual-flow network of CNN and Transformer, incorporating an attention module to fuse the extracted features.

(3) To achieve higher quality enhanced images, we employ a generative adversarial network to recover more detailed image information through image perception quality guided adversarial learning.

## RELATED WORKS

Computer technology has significant advantages in image processing, leading to a plethora of related research. *Wang (2023)* proposed a graphic design method based on visual communication technology, utilizing multi-scale Retinex to adjust global brightness and optimize visual effects of graphic design images. *Lifang & Lu (2022)* explored the application of graphic design language based on AI visual communication, focusing on the study of graphic design language. *Xin (2021)* innovated the multidimensional visual expression of graphic design elements by analyzing the current state of multidimensional visual expression and incorporating interactive design forms and network communication of graphic design elements.

Image enhancement technology can significantly improve the visual effects of images and enhance the human eye's ability to discern information (*Luna & Moustafa, 2021*; *Jahidul, Youya & Junaed, 2020*). Various methods have been developed to address this, such as an adaptive image enhancement algorithm by *Xinying & Yihui (2023)*, which

combines images degraded by dust, night, and foggy conditions for simulation verification. The visual effects of enhanced images were compared and quantitatively evaluated. *Yao et al. (2023)* proposed a nonlinear diffusion system model based on time fraction delay to enhance useful image features while suppressing irrelevant information, demonstrating the model's enhancement capabilities. To improve packaging design efficiency, *Jinping (2022)* introduced a packaging design model based on deep convolutional generative adversarial network (DCGAN) for image enhancement, which improved visual communication, image information fusion, and packaging design effectiveness.

Although these models offer advantages in image enhancement, they often target specific situations and data, facing challenges such as large number of parameters, complex model structures, and insufficient generalization. Various data forms exhibit different characteristics in data enhancement, while images frequently suffer from issues like unclear identification and low resolution, leading to low-quality levels.

To solve these problems and improve the visual effect of images in graphic design, various methods have been proposed, each with distinct strengths and weaknesses. *Paul, Bhattacharya & Maity (2022)* proposed a method of double HE based on histogram correction for contrast enhancement of digital images. The primary strength of this approach lies in its simplicity and effectiveness in improving visual quality. However, it may not perform well on images with complex lighting conditions or those requiring more nuanced enhancement techniques. *Toderici et al. (2015)* introduced an end-to-end optimized image compression method using recurrent neural networks (RNNs) to reconstruct images. The deep neural network's modeling capabilities allow for superior compression performance compared to traditional methods like JPEG and BPG. Despite its high efficiency, the complexity of training RNNs and their computational demands might limit practical applications, particularly on devices with limited resources. *Zhao, Saeid & Guillaume (2022)* tackled redundant information across different resolutions with a new region-based multi-resolution image description scheme. This method excels in converting any region-based image descriptor into a multi-resolution structure, enhancing flexibility and adaptability. Nonetheless, the complexity of managing multiple resolutions and the potential for increased computational overhead could be considered drawbacks. *Lore, Akintayo & Sarkar (2017)* presented LLNet, a pioneering low-light image enhancement algorithm based on deep learning. By employing a traditional self-codec structure with a sparse denoising autoencoder, LLNet effectively improves image contrast and reduces noise. However, its performance might degrade with extremely low-light conditions or images with significant noise, indicating a potential area for further refinement.

For underwater image enhancement, *Wang et al. (2017b)* proposed UIE-Net, a CNN-based network. UIE-Net is tailored specifically for underwater environments, addressing unique challenges like color distortion and low visibility. While it shows promising results, its specialized nature might limit its application to other types of image enhancement tasks. *Xiang et al. (2019)* introduced a method based on convolutional long short-term memory networks for low-light image enhancement. This approach benefits from the temporal dependencies modeled by long short-term memory (LSTM), offering robust performance in dynamically changing lighting conditions. However, like other deep

learning models, it requires significant computational power and extensive training data. *Zhang et al. (2023)* modified the traditional Transformer calculation method, proposing a Transformer-CNN collaborative network for enhancing low-light images. This innovative combination leverages the strengths of both Transformers and CNNs, offering improved enhancement capabilities. The complexity of integrating these two architectures, however, could pose implementation challenges. *Jiang et al. (2021)* proposed enlightened GAN, the first algorithm to effectively introduce unsupervised learning into low-light image enhancement. This GAN-based approach demonstrates remarkable adaptability and performance in varying lighting conditions. Nevertheless, GANs are notorious for their training instability and potential for generating artifacts, which could undermine their reliability in certain scenarios.

In summary, while each method presents unique advantages, they also exhibit specific limitations. Techniques like double HE and multi-resolution schemes offer simplicity and flexibility but may struggle with computational demands and complex conditions. Deep learning-based approaches, including LLNet, UIE-Net, and enlightened GAN, provide superior enhancement but often at the cost of increased computational complexity and training challenges. Future research could focus on hybrid models that combine these strengths while mitigating their respective weaknesses.

Therefore, in order to obtain higher quality image enhancement and increase its application in the field of graphic design, this paper studies the use of CNN and Transformer model to form a dual-flow network, while adding attention mechanism and using generative adversarial network for image enhancement. The method aims to extract detailed features of images, amplify local information, and improve the generalization ability of model recognition and processing to adapt to the inherent complexity and variability of graphic design.

## MODEL DESIGN

We propose a novel image enhancement algorithm that integrates a CNN-Transformer dual-flow architecture with a Generative Adversarial Network (CTGAN), as illustrated in Fig. 1. The necessity of this approach arises from the increasing demand for advanced image processing techniques capable of handling complex image data, particularly in scenarios where conventional methods fail to deliver optimal results.

The process begins with the input image undergoing HE, a vital preprocessing step designed to improve image contrast. HE redistributes the intensity values of the image across the full dynamic range, thus enhancing visibility of the details. This step is crucial as it ensures that the image features become more distinct, enabling the subsequent feature extraction processes—carried out by the CNN and Transformer networks—to work with higher-quality data. The enhanced contrast aids in capturing finer image details, ultimately improving the overall performance of the CTGAN in generating superior image outputs.

By employing this combination of techniques, the algorithm not only amplifies image clarity but also addresses issues like poor contrast, which are common challenges in image enhancement tasks.

**The first stage of image enhancement**

**Figure 1** Flowchart of the algorithm framework.

For feature extraction, the CTGAN employs a dual-flow model. In the CNN flow, the preprocessed image is passed through several convolutional layers. Each layer applies filters to extract different local features, such as edges and textures. The output of the CNN flow is a set of feature maps that highlight various local patterns in the image. Concurrently, in the Transformer flow, the same preprocessed image is fed into a Transformer. The self-attention mechanism captures long-range dependencies and global context within the image, outputting feature representations that encapsulate the overall structure and significant regions. The features extracted by both the CNN and Transformer are then combined using an attention mechanism. This mechanism assigns importance weights to the features based on their relevance, ensuring that the most critical features from both local and global contexts are highlighted.

Further enhancement is achieved using the GAN. The initially enhanced image is input into the generator network of the GAN, which refines the image by focusing on improving its perceptual quality, such as enhancing textures, colors, and overall sharpness. The generator is trained to produce images that are indistinguishable from high-quality reference images. The discriminator evaluates the images produced by the generator, comparing them to real high-quality images, and provides feedback to the generator, helping it iteratively improve the quality of its outputs.

The final output of the CTGAN model is a visually enhanced image that has undergone thorough enhancement through both local and global feature extraction, as well as perceptual refinement.

## Histogram equalization

HE is a technique that involves expanding areas with high pixel density to adjacent regions, creating layers, and gradually narrowing areas with fewer pixels. This method increases the overall gray level range, optimizes the utilization of different gray levels, and enhances image contrast (*Zhang et al., 2021*). The procedure for HE comprises the following steps:

Histogram segmentation: This step involves dividing the histogram into multiple sub-histograms, which are then processed independently. This approach aims to achieve a

more natural visual effect in the enhanced image and mitigates the problem of brightness migration commonly observed in traditional HE methods.

Histogram clipping: The primary objective of this step is to regulate the enhancement rate during the HE process and to prevent the excessive stretching of certain gray levels. This is accomplished by setting thresholds on gray levels with high and low frequencies.

Independent equalization: This step involves applying the equalization process separately to each sub-histogram. The enhancement effect may be minimal for sub-histograms with narrow ranges, whereas it may be excessive for those with broader ranges. Consequently, the mapping space for each sub-histogram is readjusted to ensure a balanced dynamic range.

HE can be popularized in discrete images. If the total number of pixels in the image is assumed to be one gray level, and the value of the first gray level is assumed to have the number of gray level pixels, gray level pixels will appear in the image (*Wu, Song & Zhang, 2020*). If the number of pixels in an image is $n$, and the gray level of the picture is $L$, then the probability of producing a fourth gray level is $m$:

$$P(r) = \frac{nm}{n} (m = 0, 1, 2, 3, 4 \dots L). \tag{1}$$

The processing function after equalization:

$$F(k) = \sum_{i=0}^{m} \frac{ni}{n}. \tag{2}$$

If the equalization range is [0-255], all gray levels in the image are mapped, yielding the following function:

$$S(K) = 255.FK \tag{3}$$

where $S(K)$ represents the scaled value of the pixel intensity $K$ after equalization; 255 represents the maximum intensity value in an 8-bit grayscale image; $F$ represents a scaling factor; $K$ represents the original intensity value before equalization.

Bilayer column balance is an optimization of histogram and single platform balance. This method prevents excess pixels (usually including background and noise) from grayscale by setting the highest platform threshold, and prevents low pixels (usually including weak objects and details) from merging through other grayscale by setting the lower platform threshold during the enhancement process. Low pixels (usually including weak objects and details) are fused by other gray levels. The calculation formula is:

$$P(k) = \begin{cases} T\max & P(k) >= T\max \\ P(k) & T\min <= P(k) <= T\max \\ T\min & 0 <= P(k) <= T\min \end{cases} \tag{4}$$

where the equation $P(k)$ represents a corrected value for the dual-platform histogram, which categorizes pixels based on their intensity values. $T_{max}$ represents the upper threshold for the pixel intensity. $T_{min}$: The lower threshold for the pixel intensity.

Equation (4) explains how pixel intensities are classified: $P(k) \geq T_{max}$—pixels with intensity values greater than or equal to the upper threshold are assigned $T_{max}$;

$T_{\min} \leq P(k) \leq T_{\max}$ - pixels within the threshold range are kept as they are; $P(k) \leq T_{\min}$—pixels with intensity values less than or equal to the lower threshold are assigned $T_{\min}$.

After modifying the statistical histogram, the cumulative histogram of the image is derived from the adjusted statistical histogram. The gray levels of the image are then redistributed using this cumulative histogram, akin to the HE method, resulting in an enhanced image with improved balance.

(1) Accumulate the revised histogram:

$$FT(k) = \sum_{i=0}^{k} \Pr(k) \, 0 < i < 25 \tag{5}$$

where $FT(k)$ is the cumulative histogram of the image.

(2) Map the gray value in the original image, taking the 8-bit image as an example:

$$Sk = 255. FT(k) \tag{6}$$

where $S_k$ is the statistical histogram of the image, $T$ is the platform threshold, and $FT(k)$ is the platform statistical histogram of the image.

## AFF based CNN-Transformer dual-stream network

Figure 2 shows the network structure diagram of the first stage of the CTGAN algorithm, which adopts a U-Net type network architecture and consists of an encoder, a decoder, and a "jump part" that provides side auxiliary input for the decoding module. The encoder adopts the two-flow feature extraction path of CNN-Transformer, and realizes feature extraction from the input image step by step through multi-level feature processing layers from bottom to top. Convolution with step 2 is used between adjacent feature processing layers to perform spatial downsampling of the feature graph. The feature processing layer of each level is determined by the residual dense block (*Hu et al., 2023*).

The encoder sequentially encodes the input image to capture diverse feature information and furnish the decoder with varying levels of local structural features and semantic details for stepwise image reconstruction. To harness the strengths of CNN in local feature extraction and Transformer in global feature capture, this paper integrates a dual-stream feature extraction pathway at the encoder's end, comprising CNN and Transformer branches.

The CNN branch takes the residual-intensive module RDB as the core module. Figure 3 shows the specific implementation of each RDB module. The essence of RDB module is a residual module with dense connections. Each basic block involved in dense connections is composed of $3 \times 3$ spatial convolution and ReLU operation.

Figure 3 illustrates the unique dense connectivity among basic blocks within the RDB, enabling each level's basic block input to integrate the output feature maps from previous levels and the input feature map of the RDB block. Through this progressively intensive connection and processing, the RDB block achieves local feature extraction of input feature maps across different levels and a multiscale receptive field perception.

In this paper, the CNN feature extraction pathway is structured around a feature extraction layer centered on the RDB block. The output feature map from the deeper

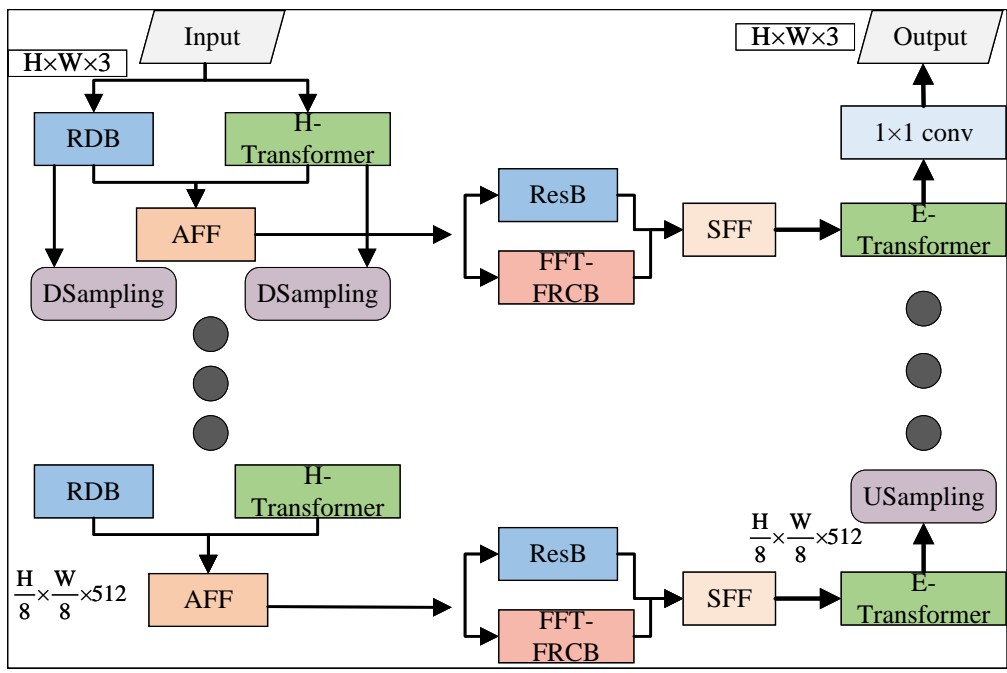

**Figure 2 Network structure diagram of the first stage of CTGAN algorithm.**

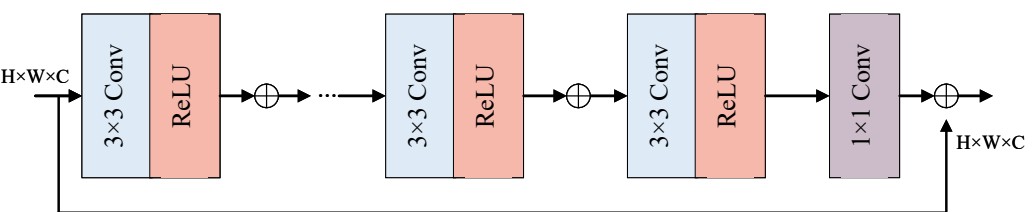

**Figure 3 Residual intensive module based on CNN.**

layers of the RDB block contains richer spatial local structural information with diverse receptive fields, built upon the subsampling of the preceding RDB block's output feature map space. Consequently, the CNN feature extraction pathway maximizes the benefits of RDB blocks for feature extraction, facilitating bottom-up local feature extraction and gradual embedding of input images.

As shown in Fig. 4, the H-Transformer consists of hierarchical multi-head self-attention (HMHSA) (*Li et al., 2023*), layer normalization (LN), and a multi-layer perceptron (MLP). The H-Transformer first divides the input feature map spatially to form a sequence of subblocks, and then applies a linear transformation *via* Patch Embedding to generate the initial feature vectors for these subblocks. This allows the feature encoding of the input feature map to be achieved using the self-attention layer, based on the normalization of subsequent layers. The context eigenvector representation of these subblock sequences is

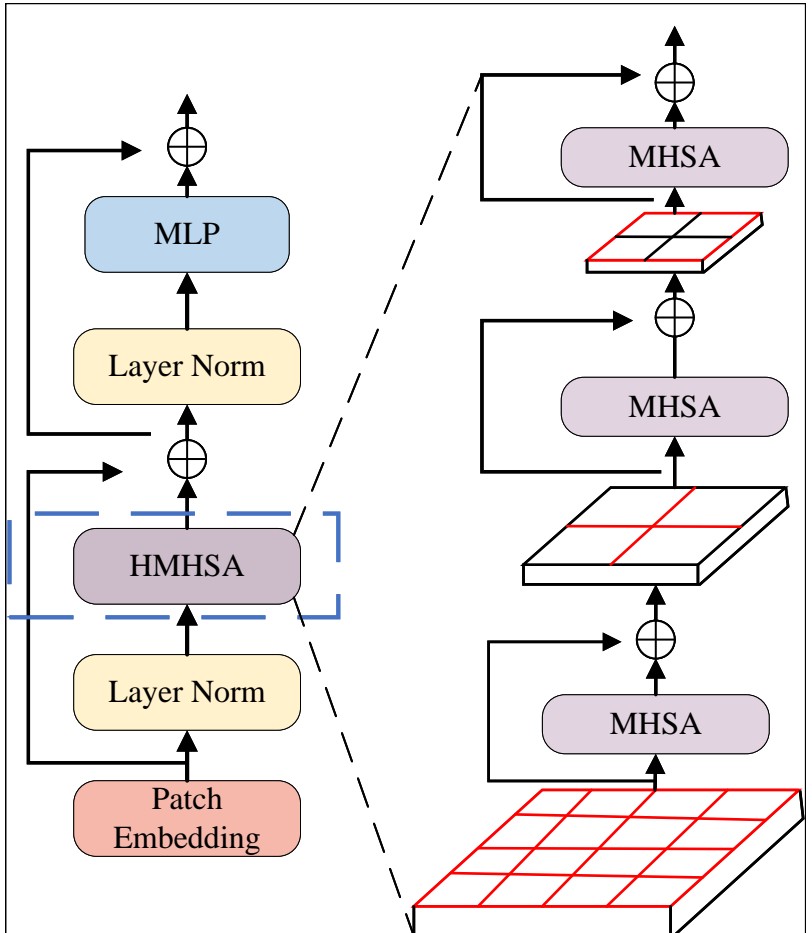

**Figure 4   Hierarchical transformer module (H-Transformer).**

then obtained, enabling the model to better capture long-range dependencies within the subblock sequences.

For the single HMHSA layer in Fig. 4, three MHSA sub-layers are cascated in a bottom-up sequence. If the space division process of the input feature map is expressed by a recursive quadtree with depth of 3, The lowest level MHSA sublayer of HMHSA pays more attention to the calculation of multi-head self-attention of each fine molecule block in the region where the leaf node is located, and therefore focuses on the modeling of spatial dependence between different fine molecule blocks in the smaller local region. Therefore, with the help of successive successive MHSA sublayers in the HMHSA layer, the progressive and nested self-attention calculation of the input feature map from the local region to the more global region is realized. Compared with the self-attention calculation of the entire feature map directly using the MHSA layer, the computational complexity is reduced.

To effectively combine the encoding features from CNN and Transformer branches, the attention feature fusion module AFF is introduced into each feature extraction layer of the encoder. As shown in Fig. 5, taking the information fusion of two branches of the

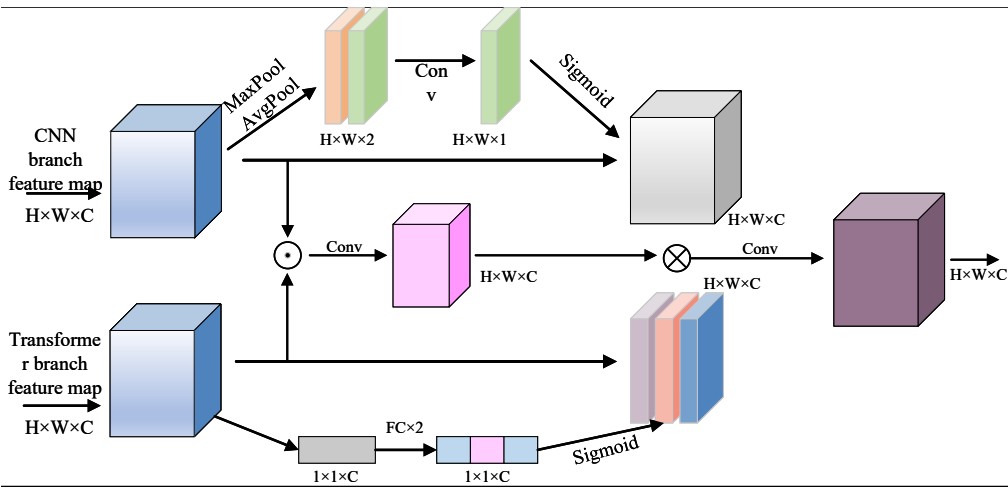

**Figure 5** Attention feature fusion module.

encoder as an example, the multi-channel feature maps of the organization feature layer of the Transformer branch and the organization feature layer of the CNN branch are simultaneously added to the AFF module. In order to enhance global information from Transformer branches, channel focus enhancement and spatial focus enhancement can be carried out, which effectively enhances spatial local structure information from CNN branches.

At the same time, using 3×3 spatial convolution, the cross-path fine-grained interaction between two sets of multi-channel feature graphs is carried out. Finally, the output of AFF module is generated by channel connection and 3×3 spatial convolution. Obviously, the AFF module skillfully integrates channel attention, spatial attention and cross-path interaction mechanism, and effectively captures the global and local context information of the input image at the current feature extraction layer with the help of dual-path feature fusion, providing effective data preparation for the "jump part" between the encoder and the decoder.

## GAN

To obtain high-quality enhanced images that align with human visual perception, this paper employs a conditional generative adversarial network (CGAN) in the second stage. By using unpaired high-visual-quality images from the Ava dataset (*Gu et al., 2018*) as input conditions, the images, which underwent initial enhancement in the first stage, undergo further colour adjustment. More detailed information is recovered through guided adversarial learning of image perceptual quality, resulting in a further enhancement of the visual effect of the initially enhanced images.

As depicted in Fig. 6, the generator adopts a U-Net architecture, which consists of an encoder, decoder, and skip connections. The encoder includes the RDB module and an undersampling layer, which is essentially a 3×3 convolutional layer with a stride of 2. The decoder comprises an RDB module and a 3×3 transposed convolutional layer with a

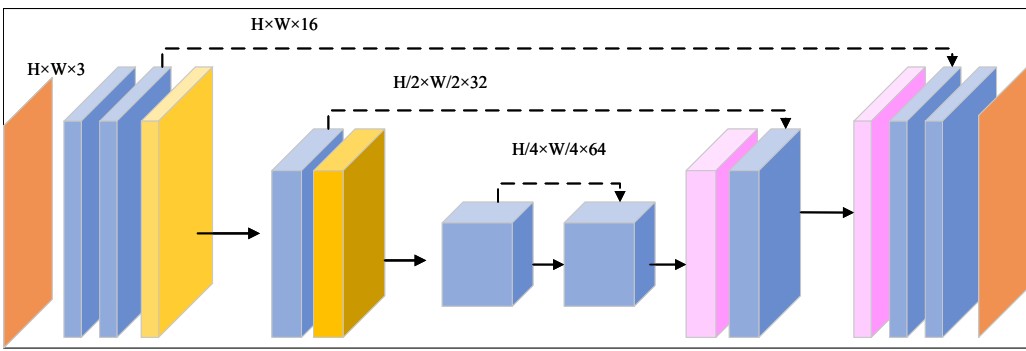

**Figure 6** Generator of the second enhancement phase of CTGAN.

stride of 2. In the skip connections, the low-level feature map from the RDB module at the encoder is laterally passed to the decoder, where it is added to the feature map of the same resolution generated by the decoding module, providing input for subsequent decoding.

Using the initially enhanced image from the first stage as input, the generator's encoder extracts local features at each layer using the RDB module, gradually abstracting input image features from low to high levels. The output of the bottleneck layer is then fed into the decoder, where the feature maps from the encoder are combined through top-down skip connections for stepwise decoding. This fusion of high-level semantic information and low-level features from the initially enhanced image leads to further image enhancement.

In the second stage of the CTGAN, the enhancement process builds upon the initially enhanced image from the first stage, rather than processing the original image directly. This approach allows the generator to achieve the desired enhancement effect with a relatively shallow U-Net architecture.

# EXPERIMENTAL RESULT AND ANALYSIS

In this section, we present the application performance of our method in graphic design based on visual characteristics, assessed through simulation experiments. We evaluate its effectiveness in comparison with established network models. Our objective is to conduct a comparative analysis of experimental results across different method types. Furthermore, this section assesses the performance of our proposed method in image enhancement through experiments. We compare the enhancement effects with those of various image enhancement methods and evaluate the accuracy of enhanced image recognition.

## Dataset and evaluation indicators

Two datasets are used in this paper, including the publicly available large-scale graphic design dataset DesignNet, linked to (https://design-net.org/, doi: https://doi.org/10.1145/3591196.3596614) and the AVA dataset, linked to (https://github.com/mtobeiyf/ava_downloader/tree/master/AVA_dataset, doi: 10.1109/CVPR.2012.6247954). The DesignNet dataset contains 4,369 visual images used in interface design, which are labeled with semantic information of style, technique, and space. Currently, each image in the dataset

**Table 1  Experimental environment.**

| Environment | Paremeter |
|---|---|
| CPU | Intel Core i9-11900F@2.50 GHz 8 cores |
| GPU | Nvidia GeForce 3060 11G |
| Operating system | Ubuntu 20.04 |
| Programming language | Python3.7 |
| Deep learning framework | pytorch |

**Table 2  Parameter settings.**

| Component | Parameter | Value |
|---|---|---|
| Preprocessing | Histogram Equalization (HE) | Applied |
| | Number of Convolutional Layers | 5 |
| CNN | Filter Sizes | $3 \times 3, 3 \times 3, 5 \times 5, 5 \times 5, 3 \times 3$ |
| Flow | Activation Function | ReLU |
| | Pooling | Max Pooling (2x2) |
| | Self-Attention Mechanism | Multi-Head Attention |
| Transformer | Number of Heads | 8 |
| Flow | Transformer Layers | 6 |
| | Positional Encoding | Sine and Cosine |
| Attention | Combination Strategy | Weighted Sum |
| Mechanism | Attention Weights | Learned via Training |
| | Generator Architecture | Deep Convolutional GAN |
| | Discriminator Architecture | Convolutional Layers |
| | Loss Function | Binary Cross-Entropy |
| GAN | Training Algorithm | Adam Optimizer |
| | Learning Rate | 0.0002 |
| | Batch Size | 16 |
| | Epochs | 150 |

corresponds to a single label in a single dimension, but in interface design practice, each design image should have the same label in three different dimensions. The AVA dataset comprises over 250,000 images, each annotated with semantic labels, style labels, and aesthetic scores. It includes 66 categories of semantic labels, 14 categories of style labels, and aesthetic scores ranging from 1 to 10. To validate the proposed graphic design image enhancement model based on visual characteristics, a series of experiments were conducted. The specific experimental settings are detailed in Table 1. Table 2 outline the model parameter settings.

This paper mainly defines several evaluation indexes for image enhancement algorithms. In addition to the common P (Precision), R (Recall) and F values, there are also two indicators: peak signal-to-noise ratio (PSNR) and structural similarity (SSIM) (*Osorio et al., 2022*; *Prodan, Vlăsceanu & Boiangiu, 2023*).

Given two images X and Y of m×n, the PSNR value of image Y relative to image X is defined as shown in Eq. (7). The unit of PSNR is dB, and the larger the value, the lower the

degree of distortion of image Y relative to image X.

$$PSNR(X,Y) = 10 \times \log 10 \left( \frac{MAX_Y^2}{MSE(X,Y)} \right) = 20 \times \log 10 \frac{MAX_Y}{\sqrt{MSE(X,Y)}} \tag{7}$$

where, $MAX_Y$ denotes the maximum pixel value in the image $Y$, which is usually taken for 256-level grayscale images, $MAX_Y = 255$;

$MSE(X,Y)$ is the mean square error of the pixel values of $X$ and $Y$ of the two images, calculated as shown in Eq. (8).

$$MSE(X,Y) = \frac{1}{m \times n} \sum_{i=0}^{m-1} \sum_{j=0}^{n-1} [X(i,j) - Y(i,y)]^2. \tag{8}$$

SSIM combines brightness, contrast and structure to measure the similarity of two images from the perspective of image composition. Given two images of m×n $X$, $Y$, calculate the SSIM between $X$ and $Y$ according to Eq. (9).

$$SSIM(X,Y) = \frac{2\mu_X \mu_Y + c_1}{\mu_X^2 + \mu_Y^2 + c_1} \times \frac{2\sigma_{XY} + c_2}{\sigma_X^2 + \sigma_Y^2 + c_2} \tag{9}$$

where $\mu$ $X$ and $Y$ are the average brightness values of $X$ and $Y$, which reflect the overall brightness of $X$ and $Y$. $\sigma$ $X$ and $Y$ are the standard deviation of brightness of $X$ and $Y$.

Compared with PSNR value, SSIM based image quality evaluation is more consistent with the visual characteristics of human eyes. SSIM ranges from 0 to 1. In the actual calculation, the given image pair is usually divided into spatial blocks, and the SSIM value of each image block is calculated at the same time. The SSIM value of the given image pair can be obtained by arithmetic average of the SSIM value of these image blocks.

## Model evaluation methods

To eliminate noise, redundancy, and inconsistencies in the data and to enhance its standardization, reliability, and applicability for specific tasks, data preprocessing is essential. For image preprocessing, we utilize mean subtraction, which involves subtracting the mean value of all training set images from each image's features during training. This process centralizes the data of each dimension to zero, reducing computational overhead. It transforms the data into a matrix composed of vectors under the original standard coordinate system, centered around the mean values of these vectors. This step can be achieved using Eq. (10):

$$X' = X - np.mean(X, axis = 0). \tag{10}$$

In this step, the mean of each feature across the entire dataset, $X$, is computed along the specified axis (in this case, axis 0, which refers to columns or features). The resulting mean vector is then subtracted from each data point in the dataset. This operation centers the data, ensuring that the average value of each feature is zero, which simplifies further computation and can enhance the performance of neural networks by normalizing input data. np.mean is a function from the Python library NumPy, which is used for numerical computing.

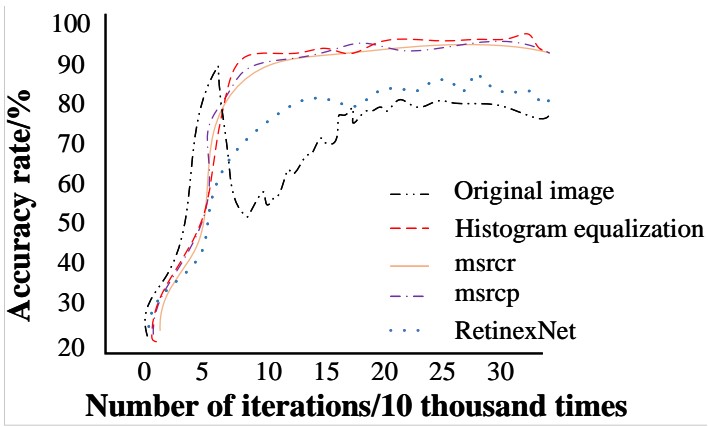

**Figure 7  Teration curves of different image enhancement methods.**

Neural network architecture was used for training, and iteration curves of different image enhancement methods were obtained, as shown in Fig. 7. As can be seen from Fig. 7, the balance was reached after about 150,000 iterations using different enhancement methods. The different enhancement algorithms used here are multi-scale Retinex with color restoration (MSRCR) (*Wang et al., 2017a*), multi-scale Retinex with chromaticity preservation (MSRCP) (*Guo et al., 2021*), and RetinexNet (*Tang et al., 2022*). MSRCR is an algorithm to improve MSR, that is, to add the function of color recovery on the basis of MSR, the expression is shown in formula Eq. (11).

$$R_{MSRCR}(x,y,\sigma) = \beta \log\left(\alpha \frac{I_t(x,y)}{\sum_{l=1}^{3} I_l(x,y)}\right) R_{MSR}(x,y,\sigma) \qquad (11)$$

where $R_{MSRCR}(x,y,\sigma)$ represents the output of the MSRCR process, which combines multi-scale Retinex (MSR) with a color restoration function.

$\beta$ and $\alpha$ represent the scaling factors used for tuning the brightness and contrast of the image. $I_s(x,y)$ represents the intensity of the image at position $(x,y)$ for the $s^{th}$ color channel. $\sum_{l=1}^{3} I_l(x,y)$ represents the sum of intensities across all three color channels (typically Red, Green, and Blue). $R_{MSR}(x,y,\sigma)$ represents the multi-scale Retinex output, which enhances the contrast and dynamic range of the image.

To address the color bias issue arising from MSRCR, the MSRCP algorithm processes the MSR on the intensity data of the image. Subsequently, it maps this data back to each channel based on the original RGB scale. In RetinexNet, the algorithm aims to mitigate or eliminate the influence of the incident light on the image by preserving the essential reflective attributes of the object. The theoretical enhancement algorithm's application formula is depicted in Eq. (12).

$$S(x,y) = R(x,y) * L(x,y). \qquad (12)$$

S(x,y) represents the final enhanced image output at pixel location (x,y). R(x,y) represents the reflectance component of the image, which is independent of illumination

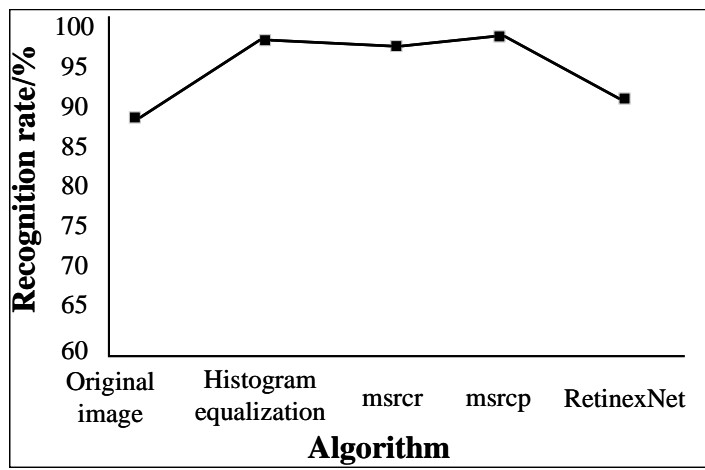

**Figure 8** Recognition success rate of different image enhancement algorithms.

and represents the intrinsic properties of the objects in the scene. L(x,y) represents the illumination component, which represents the varying light conditions in the scene.

## Results

As shown in Fig. 7, the iteration curve of the original image fluctuates greatly in the early stage, while the iteration curve based on other image enhancement algorithms is relatively smooth. In addition, the trend of HE is close to that of MSRCR and MSRCP. However, careful observation of the development trend in the early and late stages shows that the enhancement method of HE has an obvious enhancement process, and the overall recognition success rate is slightly higher than that of other image enhancement methods. Therefore, the HE enhancement method adopted in this paper has played a good role.

The average recognition success rates after the algorithm stabilizes are depicted in Fig. 8. As shown, compared to other image enhancement methods, HE achieves the highest recognition success rate, with a 14.4% increase in accuracy compared to the original image. Additionally, the recognition success rates of image enhancement algorithms using MSRCR and MSRCP are slightly lower than HE, demonstrating their effectiveness to some extent.

While the Retinex method yields a recognition success rate higher than that of the original image, the improvement is minimal and notably lower overall compared to the other three methods. Therefore, based on the final average recognition rate results, the enhanced HE exhibits robust recognition performance compared to various image enhancement algorithms.

In addition, we conducted experiments on a feature extraction method based on CNN and Transformer dual-flow models using two datasets. We compared our method with several other prominent feature models including CNN (*Munadi et al., 2020*), GoogleNet (*Chen et al., 2023*), convolutional neural network and support vector machine (CNN + SVM) (*Khairandish et al., 2022*), ResNet (*Wu, Shen & Hengel, 2019*), and Transformer (*Zhao et al., 2021*), and evaluated their performance. As depicted in Fig. 9,

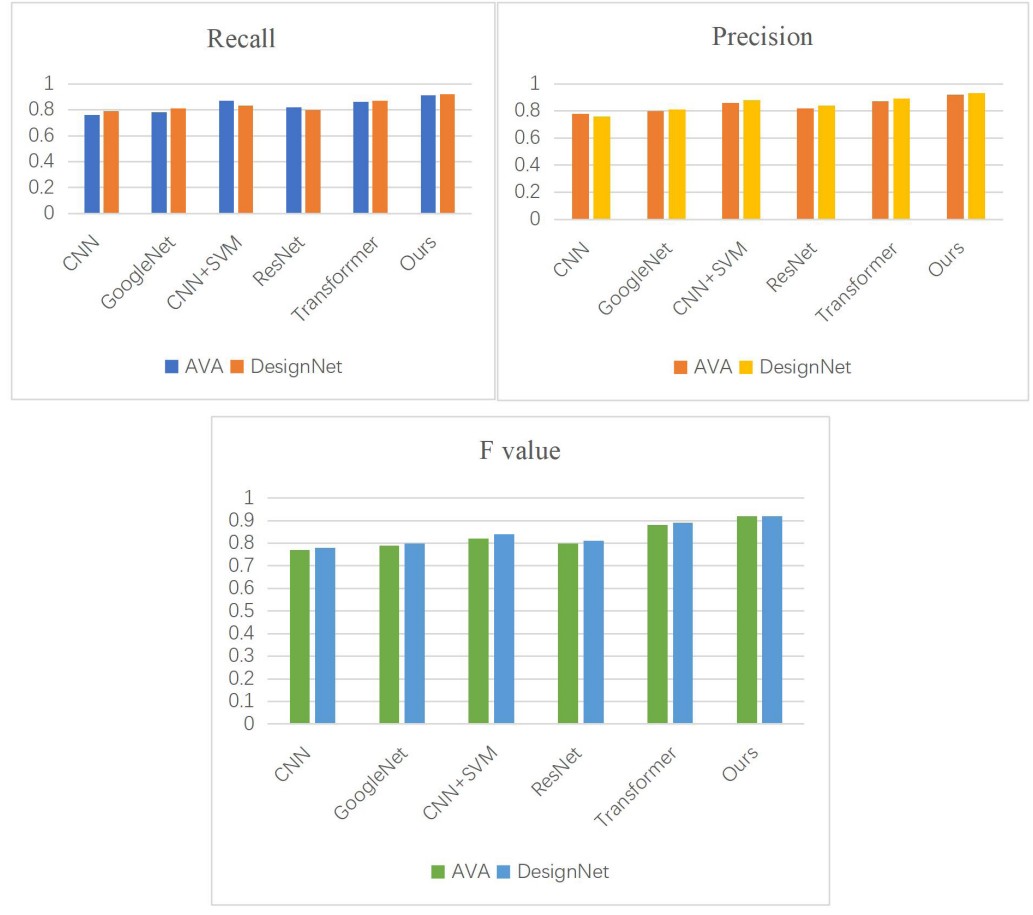

**Figure 9** Experimental results of comparison of method models.

our method achieves the highest values across all evaluation metrics compared to other algorithms.

The significant improvement in P, R and F1 score (F) values by more than 5% with our method is primarily attributed to enhancing the functionalities of CNN and Transformer, integrating them for feature extraction, and introducing an attention mechanism module to enhance feature extraction effectiveness. Consequently, our algorithm surpasses separate CNN and Transformer models by more than 10% and 5% in P, R, and F values, respectively. Compared to GoogleNet and ResNet, our approach utilizes a residual dense module at the core of the CNN branch to achieve multi-scale local feature extraction, resulting in notable improvements in P, R, and F values as observed in experimental results.

Although adding SVM improves accuracy in the CNN + SVM method compared to a standalone CNN model, our proposed algorithm consistently demonstrates stable improvements in relative accuracy across the two datasets post-experimentation.

To validate the effectiveness of the CTGAN algorithm, we conducted a comprehensive analysis by evaluating the PSNR and SSIM metrics across various algorithms using the DesignNet and AVA datasets. These metrics are critical indicators of image quality, where

**Table 3  Evaluation index results of different algorithms.** The best performance is highlighted in bold and the top performance is underlined.

| Algorithm | DesignNet dataset | | AVA dataset | |
|---|---|---|---|---|
| | **PSNR** | **SSIM** | **PSNR** | **SSIM** |
| CNN | 21.30 | 0.74 | 19.80 | 0.78 |
| CNN+SVM | 23.28 | 0.78 | 23.50 | 0.82 |
| GoogleNet | 18.27 | 0.66 | 17.85 | 0.67 |
| ResNet | 25.19 | 0.83 | 25.03 | 0.87 |
| Transformer | 24.24 | 0.85 | 25.21 | 0.88 |
| CTGAN(Ours) | **25.85** | 0.88 | **26.86** | 0.90 |

higher values denote superior performance. Table 3 presents the evaluation results, with the best performance highlighted in bold and the top performance underlined.

Quantitative comparisons reveal CTGAN's superiority in both datasets. On the DesignNet dataset, CTGAN achieved a PSNR of 25.85, surpassing ResNet (25.19) and Transformer (24.24). The improvement of 0.66 PSNR over ResNet indicates CTGAN's superior ability to reduce noise and enhance image details. On the AVA dataset, CTGAN's PSNR of 26.86 is 1.65 higher than ResNet and Transformer, highlighting its effectiveness in maintaining high-fidelity images even on diverse datasets. For SSIM, CTGAN achieved 0.88 on the DesignNet dataset, compared to ResNet's 0.83 and Transformer's 0.85, reflecting CTGAN's capability in preserving structural information and perceptual quality. On the AVA dataset, CTGAN's SSIM of 0.90 outperforms both ResNet (0.87) and Transformer (0.88), reinforcing its strength in capturing fine details and textures.

Qualitative comparisons further underscore CTGAN's advantages. Although CNN+SVM shows improved results over CNN alone, with a notable increase in PSNR and SSIM, CTGAN significantly surpasses these enhancements. The hybrid model of CNN+SVM achieves a PSNR of 23.28 and 23.50 on DesignNet and AVA, respectively, but falls short when compared to CTGAN. This indicates that while SVM aids in improving image quality, the generative approach of CTGAN offers a more substantial enhancement by generating finer details and reducing artifacts. GoogleNet's performance is the lowest among the compared models, with a PSNR of 18.27 and 17.85 and SSIM of 0.66 and 0.67 on DesignNet and AVA datasets, respectively. The traditional network architecture without residual connections limits its ability to compete with more advanced models like ResNet and CTGAN, highlighting the need for residual structures and generative adversarial frameworks for superior image quality.

ResNet and Transformer networks perform commendably, with ResNet showing strong PSNR and SSIM, and Transformer slightly better in SSIM. However, CTGAN surpasses both in all metrics, which can be attributed to the fusion of CNN and Transformer networks, coupled with the generative adversarial approach that enhances feature representation and image quality. The significant performance improvement of CTGAN demonstrates the efficacy of integrating GANs with advanced network architectures like CNN and Transformer. The generative component of CTGAN allows for better feature extraction and noise reduction, leading to higher fidelity in the reconstructed images. Moreover, the

**Table 4  Ablation results.** The best results are shown in bold.

| Method | Ablation study results | | AVA Dataset | |
|---|---|---|---|---|
| | PSNR (dB) | SSIM | PSNR (dB) | SSIM |
| Original Image (Baseline) | 20.45 | 0.650 | 21.10 | 0.680 |
| HE Only | 23.30 | 0.720 | 23.00 | 0.730 |
| CNN Flow | 24.15 | 0.750 | 25.05 | 0.760 |
| Transformer Flow | 23.85 | 0.735 | 24.75 | 0.745 |
| CNN + Transformer | 25.40 | 0.765 | 26.55 | 0.780 |
| CTGAN (Full Model) | **25.85** | 0.88 | **26.86** | 0.90 |

adversarial training helps in refining details that are crucial for both quantitative metrics and visual quality.

The ablation study results are illustrated in Table 4.

Original image (Baseline): The image without any processing.

HE only: Image processed with histogram equalization only.

CNN flow: Image enhanced using only CNN feature extraction.

Transformer flow: Image enhanced using only Transformer feature extraction.

CNN + Transformer: Combination of CNN and Transformer for feature extraction and enhancement.

CTGAN (Full Model): Full CTGAN model, including HE processing, CNN flow, Transformer flow, and GAN generator.

The results of the ablation study provide critical insights into the efficacy of various image enhancement strategies. The baseline image, which is left unprocessed, serves as a control, offering a moderate level of quality, evidenced by a PSNR of 20.45 dB and an SSIM of 0.650. This establishes a fundamental benchmark against which enhancement techniques are evaluated. The use of HE alone substantially improves image quality, as reflected by notable increases in both PSNR and SSIM scores. These improvements suggest that HE effectively enhances contrast and detail visibility, though it falls short of achieving the superior metrics delivered by more sophisticated methods.

When CNNs are incorporated into the enhancement process, a further elevation in image quality is observed. The increase in PSNR and SSIM from the HE-only method to the CNN-enhanced approach implies that CNNs contribute crucial features that improve detail and clarity. This aligns with the well-established role of CNNs in identifying and amplifying intricate patterns within images. Similarly, the application of Transformer-based feature extraction results in comparable improvements to those achieved with CNNs, albeit with slight variations in performance metrics. This suggests that Transformers, known for their capacity to capture long-range dependencies and contextual information, also contribute positively to image quality. However, the performance of the Transformer-only approach does not exceed that of the CNN-based method.

The combination of CNN and Transformer techniques yields the most significant improvements, demonstrating that the integration of these methods harnesses their respective strengths in feature extraction. This joint approach capitalises on the CNN's

ability to extract detailed features and the Transformer's capability for contextual understanding, leading to the highest observed PSNR and SSIM values. Ultimately, the full CTGAN model, which integrates histogram equalization, CNN and Transformer feature extraction, and GAN-based generation, achieves the best performance across all metrics. This comprehensive strategy underscores the synergetic effects of combining multiple advanced techniques, indicating that the CTGAN model not only enhances image quality through individual stages but also refines the final output *via* generative adversarial methods.

## DISCUSSION

Under the backdrop of applying image processing technology in graphic design, this paper introduces the CTGAN algorithm, which notably enhances the effectiveness of graphic design image enhancement, leading to improved recognition, classification, and application outcomes. This model algorithm surpasses existing common methods such as Transformer and CNN +SVM (*Khairandish et al., 2022*). Notably, CTGAN excels over Transformer in P, R, and F values, primarily due to its integration of attention mechanisms like AFF to capture finer local details. Moreover, our algorithm maintains lower network complexity compared to multi-model integration methods, highlighting the importance of robust model fusion algorithms in this research field. The CTGAN model described in the paper leverages fusion, attention mechanisms, and generative networks to enhance the usability of graphic design images, thereby fostering growth in the design industry. Its superiority over other models in real dataset tests underscores its efficacy.

As contemporary visual graphic design continues to evolve, it becomes crucial to deeply understand current technological and design landscapes. With technological advancements, graphic design becomes more diverse, and integrating image enhancement technology based on visual features opens up new creative possibilities and innovative directions for artists and designers. This approach enhances the value and artistic creation potential of graphic design methods across different design images. Furthermore, advancing graphic design levels can lead to the development of various offline cultural and creative products, harmoniously integrating intelligent technology with cultural products. This enhances personalization, visual impact, and user experience, introducing novel expressions for future product design.

While AI technologies propel the growth of graphic-related fields, the future of graphic design presents both opportunities and challenges. Designers can leverage AI's supportive role alongside their creativity to achieve superior design outcomes. They must also cultivate a learning mindset to adapt to emerging technologies and continuously elevate their design proficiency in response to evolving design environments.

## CONCLUSION

This paper introduces an image enhancement method that combines HE for initial image processing and feature extraction using a CNN and Transformer dual-flow network, specifically tailored for applications in graphic design. The model algorithm initially

applies HE to process images, followed by extracting image features using the dual-flow network model. By integrating the attention module AFF, semantic features from both network models are deeply fused to enhance feature utilization. Subsequently, a generative network is employed to capture more localized detailed features, thereby improving enhancement and classification accuracy. The proposed algorithm undergoes testing on two datasets, demonstrating significant improvements over other algorithms with PRF values exceeding 90%. Additionally, evaluation indices such as PSNR and SSIM also show strong performance, highlighting the model's efficacy for future graphic design images and classification tasks. Future research aims to enhance the generalization capabilities of current models, making them applicable to a broader range of images. Furthermore, while the algorithm in this paper excels in image enhancement, there remains scope for enhancing its applicability to hand-drawn and other cultural works.

## ACKNOWLEDGEMENTS

We thank the anonymous reviewers whose comments and suggestions helped to improve the manuscript.

### Funding

This article is a National Social Science Fund project "Research on the Protection and Inheritance of Yao's' Ancient Legal Drum 'Traditional Skills'" (No. 21BMZ037). The funders had no role in study design, data collection and analysis, decision to publish, or preparation of the manuscript.

### Grant Disclosures

The following grant information was disclosed by the authors:
National Social Science Fund project "Research on the Protection and Inheritance of Yao's' Ancient Legal Drum 'Traditional Skills'": 21BMZ037.

### Competing Interests

The authors declare there are no competing interests.

### Author Contributions

- Ming Xu conceived and designed the experiments, analyzed the data, authored or reviewed drafts of the article, and approved the final draft.
- Jinwei Cui conceived and designed the experiments, prepared figures and/or tables, and approved the final draft.
- Xiaoyu Ma performed the experiments, performed the computation work, authored or reviewed drafts of the article, and approved the final draft.
- Zhiyi Zou performed the experiments, prepared figures and/or tables, authored or reviewed drafts of the article, and approved the final draft.
- Zhisheng Xin conceived and designed the experiments, analyzed the data, performed the computation work, prepared figures and/or tables, and approved the final draft.

- Muhammad Bilal conceived and designed the experiments, performed the experiments, performed the computation work, prepared figures and/or tables, and approved the final draft.

## Data Availability

The raw measurements are available in the Supplementary File.

The data is available at:

- DesignNet: https://design-net.org.
- AVA: https://huggingface.co/datasets/Iceclear/AVA/tree/main.

## Supplemental Information

Supplemental information for this article can be found online at http://dx.doi.org/10.7717/peerj-cs.2417#supplemental-information.

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
