# Peer review of "Image enhancement with art design: a visual feature approach with a CNN-transformer fusion model"

_PeerJ Computer Science, doi:10.7717/peerj-cs.2417_

## Round 0.1 · original submission · Major Revisions

Please revise the paper by incorporating comments from all the reviewers and resubmit for further consideration.

**Language Note:** The review process has identified that the English language must be improved. PeerJ can provide language editing services - please contact us at [email protected] for pricing (be sure to provide your manuscript number and title). Alternatively, you should make your own arrangements to improve the language quality and provide details in your response letter. – PeerJ Staff

·

Basic reporting

The paper is really interesting with an high level scientific sound. It perfectly fits with the main aim of the Journal. I have carefully checked the mathematical and computational framework working so well. I have found it robust and accurate so well. The Literature references are complete and reflects the contents citations.

Experimental design

It is well

Validity of the findings

The findings are robust and accurate and really increase the knowledge in the related literature

Additional comments

none

Reviewer 2 ·

Basic reporting

The manuscript entitled “Image Enhancement with Art Design: A Visual Feature Approach with a CNN-Transformer Fusion Model” has been investigated in detail. This paper presents a novel image enhancement model for graphic design images, incorporating advanced techniques such as histogram equalization (HE), a dual-flow network with CNN and Transformer architectures, and an improved attention mechanism module. While the approach is innovative and relevant, several critical issues need to be addressed to enhance the clarity, depth, and impact of the paper. The following points highlight areas that require revision and improvement.
1) The introduction lacks specific details about the unique contributions of this model. Clearly define the problem being addressed and the specific gaps in current solutions that this model aims to fill.
2) Provide more context about the importance of image enhancement in graphic design, including real-world applications and challenges. Explain why the proposed methods (CNN, Transformer, AFF) are particularly suited for this task.
3) The literature review is not exhaustive. Ensure that all significant and recent studies in image enhancement, especially in the context of graphic design, are included.
4) The review lacks critical analysis. Discuss the strengths and weaknesses of different approaches, and compare them systematically to highlight the novelty and advantages of your proposed model.

Experimental design

5) The descriptions of the methods used (HE, CNN, Transformer, RDB, AFF) are too superficial. Provide a more detailed explanation of these techniques and how they are integrated into your model.
6) Include more technical details about the implementation. Use diagrams or flowcharts to illustrate the architecture of the proposed model and the workflow of the image enhancement process.
7) Provide more details about the experimental setup, including the datasets used, the training process, hyperparameters, and the hardware configuration. This information is crucial for reproducibility.

Validity of the findings

8) Describe the two large image datasets used for evaluation in more detail, including their sources, size, and characteristics.
9) The authors should clearly emphasize the contribution of the study. Please note that the up-to-date of references will contribute to the up-to-date of your manuscript. The studies named- “Overcoming nonlinear dynamics in diabetic retinopathy classification: a robust AI-based model with chaotic swarm intelligence optimization and recurrent long short-term memory; Artificial intelligence-based robust hybrid algorithm design and implementation for real-time detection of plant diseases in agricultural environments”- can be used to explain the methodology in the study or to indicate the contribution in the Introduction section.
10) The discussion of results is limited. Provide a more in-depth analysis of the experimental results, including quantitative and qualitative comparisons with baseline models.
11) Evaluation Metrics: Explain the significance of PSNR and SSIM metrics in the context of image quality evaluation. Discuss how the proposed model improves these metrics and what this means for practical applications.

Reviewer 3 ·

Basic reporting

The authors proposed image enhancement technique using CNN model.

Experimental design

Few quantitative measures are given in the experimental analysis using image datasets.

Validity of the findings

More findings should be done.

Additional comments

Comments are given below:
1. Authors should define every variable for different equations.
2. In equation 1, the range of the variable "L" is not defined.
3. More quantitative measures should be incorporated into quantitative comparisons.
4. More image datasets to be included.
5. No such image comparison is shown.
6. More state-of-the-art algorithms should be included in the comparison analysis. (At least 4 latest methods, not earlier than 2019).
7. The English language should be checked.

Reviewer 4 ·

Basic reporting

Very poor and unclear like "HE involves expanding areas with a high pixel density to adjacent regions, creating layering, and gradually narrowing areas with fewer pixels." It seems the authors do not have a clear understanding of the concepts.

Experimental design

Again a very poor experimental design, for example, authors are using so many things like CNN, Transformers, UNET, GANs and many others, but no intuition or justification is given for any. Moreover, the authors are using such an advanced DL architecture, then HE is at all required. No ablation study is performed. The presentation of results and analysis is also vague.

Validity of the findings

Unsatisfactory

Additional comments

NA

---

## Round 0.2 · Minor Revisions

Dear Authors, the reviewers still have concerns specially related to the paper presentation and definition/refences of some terms used in the manuscript. Please look into all of these minor issues and after addressing you may re-submit for further evaluation.

Reviewer 2 ·

Basic reporting

It is acceptable in the present form.

Experimental design

It is acceptable in the present form.

Validity of the findings

It is acceptable in the present form.

Reviewer 3 ·

Basic reporting

This is a type of image enhancement method

Experimental design

OK

Validity of the findings

POOR

Additional comments

Comments are given below:
1. Still authors could not define variables, clearly (Eq. 4). Equations are not written clearly (Eq. 3) and many more.

2. Freely available image datasets are many. Authors could use those with proper citation(s). Visual evaluations are missing.

3. Authors could not cite proper citations. Example, when definitions of quantitative metrics are given authors could not give proper reference(s) of it.

Reviewer 5 ·

Basic reporting

There are some points that need to improve for clarity and academic rigor. The quality of the work is good, and the authors effectively present details of the conducted research. However, further improvements are needed before publication. Specifically:
1) The paper's structure is well managed, but further enhancements are necessary for reader comprehension.
2) There are still grammatical improvements needed, and the authors should aim for a more academic writing style.
3) Make sure the heading's first letter is an upper letter.

Experimental design

The experimental design is structured well, however further improvements require in model design phase to ensure the paper quality.

In section 3, The image enhancement is introduced abruptly; a smoother explaining its necessity and application would enhance comprehension. Further, explain the attention mechanism with details.

Validity of the findings

In result section, the evaluation metrics such as P, R, F values used to indicate the model performance, it could be more improve my specifying the exact improvement in numbers over the other study. Additional, Equation 10 need to be clear for reader, what it mean like X and np(Numpy).

Additional comments

Ablation study presents an interesting information but lacks depth in explanation.

---

## Round 0.3 · accepted · Accept

Thank you for submitting the revised version. The minor revisions has been incorporated by authors and satisfactory.